# ALIGNING TEXT-TO-IMAGE DIFFUSION MODELS WITH REWARD BACKPROPAGATION

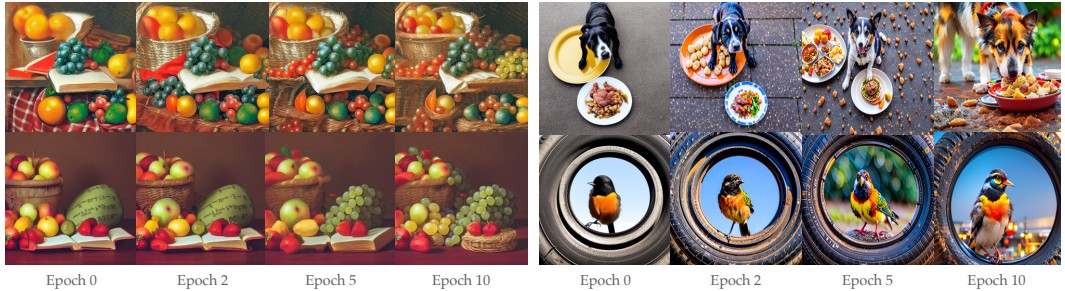

Figure 1: We present a direct backpropagation-based approach to adapt text-to-image diffusion models for desired reward function. The above examples showcase the adaptation of diffusion model output (epoch 0) through a sequence of adaptation steps (epoch 1-10) to different reward functions. The reward function in the left two examples is that of *concept removal* trained to disregard the concept of "books," despite the prompt explicitly mentioning "fruits and books." The reward function for the adaptation on the right is that of *human-preference alignment* crafted from human rankings of image-text pairs. As shown in all the examples, the proposed approach can effectively align the diffusion model with the reward function.

## ABSTRACT

Text-to-image diffusion models have recently emerged at the forefront of image generation, powered by very large-scale unsupervised or weakly supervised text-to-image training datasets. Due to their unsupervised training, controlling their behavior in downstream tasks, such as maximizing human-perceived image quality, image-text alignment, or ethical image generation, is difficult. Recent works fine-tune diffusion models to downstream reward functions using vanilla reinforcement learning, notorious for the high variance of the gradient estimators. In this paper, we propose AlignProp, a method that aligns diffusion models to downstream reward functions using end-to-end backpropagation of the reward gradient through the denoising process. While naive implementation of such backpropagation would require prohibitive memory resources for storing the partial derivatives of modern text-to-image models, AlignProp finetunes low-rank adapter weight modules and uses gradient checkpointing, to render its memory usage viable. We test AlignProp in finetuning diffusion models to various objectives, such as image-text semantic alignment, aesthetics, compressibility and controllability of the number of objects present, as well as their combinations. We show AlignProp achieves higher rewards in fewer training steps than alternatives, while being conceptually simpler, making it a straightforward choice for optimizing diffusion models for differentiable reward functions of interest.

## 1 INTRODUCTION

Diffusion probabilistic models (Sohl-Dickstein et al., 2015; Goyal et al., 2017; Ho et al., 2020a) are currently the de facto standard for generative modeling in continuous domains. Text-to-image diffusion models such as DALLE (Ramesh et al., 2022), Imagen (Saharia et al., 2022), and Latent Diffusion (Rombach et al., 2022) are at the forefront of image generation by training on web-scale data. However, most use cases of diffusion models are related to downstream objectives such as

aesthetics, fairness, text-to-image alignment, or robot task achievement, which may be not well aligned with maximizing likelihood in the training dataset. For example, while the training images may contain unusual camera and object viewpoints with objects half-visible or truncated at the image border, human users usually prefer image samples of standardized viewpoints, beautifully centered content with key elements in focus. Additionally, because of the noise present in pre-training datasets, there is frequently a misalignment between the semantics of the generated images and the associated text prompts. This occurs because the models tend to adopt the noise, biases, and peculiarities inherent in the training data. In this paper, we consider the problem of training diffusion models to optimize downstream objectives directly, as opposed to matching a data distribution.

The most straightforward approach to aligning pre-trained models to downstream objectives is supervised fine-tuning on a small-scale human-curated dataset of high-quality model responses (Ouyang et al., 2022; Lee et al., 2023). Unfortunately, this is mostly not a feasible option. It is not only difficult to collect data on samples that display desired properties like aesthetics, fairness, and text-to-image alignment, but such data can easily be biased. On the other hand, it is much easier to ask humans for relative feedback by showing two or more samples. Hence, as a result, the common practice is to train a reward model by explicitly collecting data of human preferences by asking a human subject to rank a number of examples as per the desired metric. However, in the case of diffusion models, this leads to a unique challenge: given such a reward function, how does one update the weights of the diffusion model?

The core architecture of diffusion models operates by iteratively refining a data sample through a sequence of stochastic transformations. Even though the learned reward function is differentiable, it is non-trivial to update the diffusion model through the long chain of diffusion sampling as it would require prohibitive memory resources to store the partial derivatives of all neural layers and denoising steps. This can easily be on the order of several terabytes of GPU memory (Wallace et al., 2023) for the scale of modern text-to-image diffusion models. As a result, the typical alternative is to use reinforcement learning and directly update the diffusion weights via REINFORCE. This is the most common approach today to align diffusion models with a reward function (Black et al., 2023; Lee et al., 2023; Ziegler et al., 2020; Stiennon et al., 2020). However, RL methods are notorious for high variance gradients and hence often result in poor sample efficiency.

In this paper, we revisit the idea of direct end-to-end backpropagation of the gradients of a differentiable reward function through the diffusion chain, and device practical ways to get around the issue of using exponential memory and compute. We propose Alignment by Backpropagation (**AlignProp**), a model that casts denoising inference of text-to-image diffusion models as a differentiable recurrent policy. This policy effectively maps conditioning input prompts and sampled noise to output images, and fine-tunes the weights of the denoising model using end-to-end backpropagation through differentiable reward functions applied to the output-generated image.

We fine-tune low-rank adapter weights (Hu et al., 2021), added to the original denoising U-Net, instead of the original weights, and we use gradient checkpointing (Gruslys et al., 2016; Chen et al., 2016) to compute partial derivatives on demand, as opposed to storing them all at once. In this way, AlignProp incurs reasonable memory cost while only doubling the processing cost per training step, which gets compensated due to the fact that direct backdrop needs less number of steps to optimize. However, end-to-end backdrop quickly tends to over-optimize the model to excessively maximize the reward model leading to collapse. We address the over-optimization (Gao et al., 2022) with randomized truncated backpropagation (Tallec & Ollivier, 2017), i.e., randomly sampling the denoising step up to which we back-propagate the reward.

We test AlignProp in finetuning StableDiffusion (Rombach et al., 2022) to maximize objectives of aesthetic quality, text-to-image semantic alignment, and modulating object presence, as we show in Figure 1. We show it achieves higher rewards and is more preferred by human users than reinforcement learning alternatives of (Black et al., 2023; Lee et al., 2023). We ablate design choices of the proposed model and show the importance of backpropagating through the denoising chain for variable number of steps. We show adapted layers in early denoising steps align the semantic content while adapted layers in later denoising steps adapt the high frequency details to the downstream objective. Last, we show convex combinations of weights of the finetuned models maximize combinations of the corresponding reward functions.

## 2 RELATED WORK

Denoising diffusion models (Sohl-Dickstein et al., 2015; Goyal et al., 2017; Ho et al., 2020a) have emerged as an effective class of generative models for modalities including images (Ramesh et al., 2021; Rombach et al., 2022; Saharia et al., 2022), videos (Singer et al., 2022; Ho et al., 2022a;b), 3D shapes (Zeng et al., 2022) and robot or vehicle trajectories (Ajay et al., 2023; Pearce et al., 2023; Chi et al., 2023; Tan et al., 2023). Remarkably, they have also been adapted for text generation (Lovelace et al., 2022; Lin et al., 2023) and have demonstrated utility in discriminative tasks like Image Classification (Li et al., 2023; Prabhudesai et al., 2023). Diffusion models are often pre-trained in large usupervised or very weakly supervised datasets, and adapted to improve their performance in downstream tasks, as well as their alignment with user intent. Some forms of adaptation do not alter the parameters of the diffusion model. Instead, they optimize conditional input prompts (Hao et al., 2022; Gal et al., 2022; Kumari et al., 2023), manipulate the text-image cross attention layers (Feng et al., 2023) to improve text-to-image alignment, guide sampling during inference using the gradients of a pre-trained classifier (Dhariwal & Nichol, 2021), or use classifier-free (Ho & Salimans, 2021) guidance by combining conditional and unconditional diffusion models. Other adaptation methods finetune the model's parameters using small-scale human-curated datasets, or human labels of absolute or relative quality of model's responses (Lee et al., 2023; Black et al., 2023; Wu et al., 2023b; Dong et al., 2023). These methods first fit a neural network reward function using human scores or relative preferences, and then finetune the diffusion model using reward-weighted likelihood (Lee et al., 2023), reward-filtered likelihood (Dong et al., 2023) or reinforcement learning (Black et al., 2023), such as PPO (Schulman et al., 2017). In this paper, we show how diffusion models can be finetuned directly for downstream differentiable reward models using end-to-end backpropagation. Our formulation resembles deep deterministic policy gradients (Silver et al., 2014), a reinforcement learning method that trains a deterministic policy through end-to-end differentiation of a concurrently trained Q function. Work of (Xu et al., 2023) also consider backpropagating through a differentiable reward function but only backpropagates through a single step of the sampling process. We show that AlignProp outperforms it by backpropagating gradients through the entire sampling chain. Work of (Wallace et al., 2023) also backpropagates through the sampling process of the diffusion chain but instead of optimizing the weights of the diffusion model, it optimizes the starting noise. This process is expensive, as it needs to be performed for each prompt separately.

## 3 BACKGROUND

We first discuss background material in Section 3 and present details of our method in Section 4.

**Diffusion Models** A diffusion model learns to model a probability distribution $p(x)$ by inverting a process that gradually adds noise to a sample $x$, which is known as forward diffusion process. The amount of noise added varies according to the diffusion timestep $t$ following a variance schedule of $\{\beta_t \in (0,1)\}_{t=1}^T$. Forward diffusion process adds noise to $x$ following $x_t = \sqrt{\bar{\alpha}_t}x + \sqrt{1 - \bar{\alpha}_t}\epsilon$ where $\epsilon \sim \mathcal{N}(\mathbf{0}, \mathbf{1})$, is a sample from a Gaussian distribution (with the same dimensionality as $x$), $\alpha_t = 1 - \beta_t$, and $\bar{\alpha}_t = \prod_{i=1}^t \alpha_i$. It then learns a reverse diffusion process that predicts the noise $\hat{\epsilon}$ given $x_t$ and $t$ as input The denoising process is modeled by a neural network $\hat{\epsilon} = \epsilon_\theta(x_t; t)$ that takes as input the noisy sample $x_t$ and the noise level $t$ and tries to predict the noise component $\epsilon$.

Diffusion models can be easily extended to model $p(x|\mathbf{c})$, where $\mathbf{c}$ is the conditioning signal, such as image caption, image category or semantic maps etc. . This is done by adding an additional input to the denoising neural network $\epsilon_\theta$. For this work we consider text-conditioned image diffusion models such as Stable Diffusion (Rombach et al., 2022), which are trained using a large collection of image-text pairs $\mathcal{D}' = \{(x^i, \mathbf{c}^i)\}_{i=1}^N$ using the following objective:

$$\mathcal{L}_{\text{diff}}(\theta; \mathcal{D}') = \frac{1}{|\mathcal{D}'|} \sum_{x^i, \mathbf{c}^i \in \mathcal{D}'} ||\epsilon_\theta(\sqrt{\bar{\alpha}_t}x^i + \sqrt{1 - \bar{\alpha}_t}\epsilon, \mathbf{c}^i, t) - \epsilon||^2.$$

This loss corresponds to a reweighted form of the variational lower bound for $\log p(x|\mathbf{c})$ (Ho et al., 2020b).

In order to draw a sample from the learned distribution $p_\theta(x|\mathbf{c})$, we start by drawing a sample $x_T \sim \mathcal{N}(\mathbf{0}, \mathbf{1})$. Then, we progressively denoise the sample by iterated application of $\epsilon_\theta$ according to a specified sampling schedule (Ho et al., 2020b; Song et al., 2020), which terminates with $x_0$ sampled from $p_\theta(x)$:

$$x_{t-1} = \frac{1}{\sqrt{a_t}} \left( x_t - \frac{\beta_t}{\sqrt{1 - \bar{\alpha}_t}} \epsilon_\theta(x_t, t, \mathbf{c}) \right) + \sigma_t \mathbf{z}, \text{ where } \mathbf{z} \sim \mathcal{N}(\mathbf{0}, \mathbf{1})$$

In denoising diffusion implicit models (DDIMs) (Song et al., 2022), there is no added noise in intermediate steps, and the only stochasticity in sampling comes from the initially sampled noise $x_T \sim \mathcal{N}(\mathbf{0}, \mathbf{1})$:

$$x_{t-1} = \frac{1}{\sqrt{a_t}} \left( x_t - \frac{\beta_t}{\sqrt{1 - \bar{\alpha}_t}} \epsilon_\theta(x_t, t, \mathbf{c}) \right). \tag{1}$$

## 4 ALIGNPROP

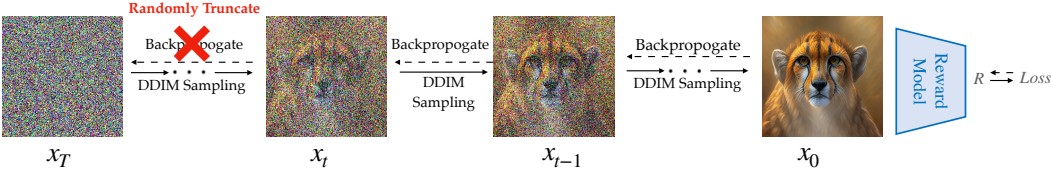

Figure 2: Given a batch of prompts, denoted as $\mathcal{P}$, AlignProp generates images from noise $x_T$ through DDIM Sampling. Subsequently, these generated images undergo evaluation by the Reward model $R_\phi$ to acquire a corresponding reward score. The optimization process involves updating the weights in the diffusion process by minimizing the negative of the obtained reward through gradient descent. To mitigate overfitting, we randomize the number of time-steps we backpropagate gradients to.

The architecture for AlignProp is illustrated in Figure 2 We introduce a method that transforms denoising inference within text-to-image diffusion models into a differentiable recurrent policy, which adeptly correlates conditioning input prompts and sampled noise to produce output images. This approach facilitates fine-tuning of the denoising model's weights through end-to-end backpropagation, guided by differentiable reward functions applied to the generated output image.

The proposed model casts conditional image denoising as a single step MDP with states $\mathcal{S} = \{(x_T, \mathbf{c}), x_T \sim \mathcal{N}(\mathbf{0}, \mathbf{1})\}$, actions are the generated image samples, and the whole DDIM denoising chain of Eq. 1 corresponds to a differentiable policy that maps states to image samples: $\mathcal{A} = \{x_0 : x_0 \sim \pi_\theta(\cdot|x_T, \mathbf{c}), x_T \sim \mathcal{N}(\mathbf{0}, \mathbf{1})\}$. The reward function is a differentiable function of parameters $\phi$ that depends only on generated images $R_\phi(x_0), x_0 \in \mathcal{A}$. Given a dataset of prompts input $\mathcal{P}$, our loss function reads:

$$\mathcal{L}_{\text{align}}(\theta; \mathcal{P}) = -\frac{1}{|\mathcal{P}|} \sum_{\mathbf{c}^i \in \mathcal{P}} R_\phi(\pi_\theta(x_T, \mathbf{c}^i)) \tag{2}$$

We update the parameters of the diffusion model using gradient descent on $\mathcal{L}_{\text{align}}$. The policy $\pi$ is recurrent and training it is reminiscent of backpropagation through time used for training recurrent networks. The gradient through update the parameters of diffusion model w.r.t downstream object (i.e., differentiable reward function) looks like:

$$\nabla_\theta \mathcal{L}_{\text{align}} = \frac{\partial \mathcal{L}_{\text{align}}}{\partial \theta} + \underbrace{\sum_{t=0}^{T} \frac{\partial \mathcal{L}_{\text{align}}}{\partial x_t} \cdot \frac{\partial x_t}{\partial \theta}}_{\text{"memory inefficient"}}. \tag{3}$$

### 4.1 REDUCING MEMORY OVERHEAD

Naively optimizing Eq. 2 with backpropagation requires to store the intermediate activations for each neural layer and denoising timestep of $\pi_\theta$ in the GPU VRAM. As can be seen in Eq. 1, the number

of activations to store for $\pi_\theta$ scale linearly with the number of diffusion timesteps. Therefore, our alignment finetuning would cost $T$ times more memory than training the diffusion model $\epsilon_\theta$ with diffusion loss of Eq. 1. For instance, training StableDiffusion (Rombach et al., 2022) using a batch size of 1 takes about 20GBs of GPU RAM, therefore training our policy $\pi_\theta$ comprised of T chained denoising models with end-to-end backpropagation would require about 1TB of GPU RAM, which is infeasible. We use two design choice to enable full backpropagation through the denoising chain: **1.** Finetuning low-rank adapter (LoRA) modules (Hu et al., 2021) in place of the original diffusion weights, and **2.** Gradient checkpointing for computing partial derivatives on demand (Gruslys et al., 2016; Chen et al., 2016).

**Finetuning LoRA weights:** Instead of fine-tuning the weights $\theta$ of the original diffusion model $\epsilon_\theta$, we add low-rank weight kernels in each of its linear neural layers, and only finetune these weights, as proposed in (Hu et al., 2021). Specifically, each linear layer of the Unet of StableDiffusion is modified from $h = Wx$ into $h = Wx + BAx$, where $W \in \mathbb{R}^{m \times m}, A \in \mathbb{R}^{m \times k}, B \in \mathbb{R}^{k \times m}$, where $k << m$. LoRA weights are initialized at 0 and do not affect the performance of the pre-trained model initially. Effectively, this means we finetune $800K$ parameters instead of $800M$, which reduces our GPU RAM usage by 2X to about 500GBs.

**Gradient Checkpointing:** Gradient checkpointing is a well known technique used to reduce the memory footprint of training neural networks (Gruslys et al., 2016; Chen et al., 2016). Instead of storing all intermediate activations in memory for backpropagation, we only store a subset and recompute the rest on-the-fly during the backward pass. This allows for training deeper networks with limited memory at the cost of increased computation time. We find that gradient checkpointing significantly reduces our memory usage from 512 GBs to 15GBs, thus making it feasible to do full backpropogation on a single GPU.

| All LoRA Disable | 0-10 timesteps | 10-20 timesteps | 20-30 timesteps | 30-40 timesteps | 40-50 timesteps | No LoRA Disable |

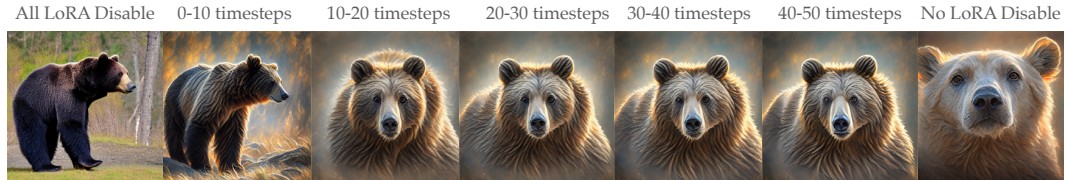

Figure 3: We illustrate the impact of deactivating finetuned LoRA weights across varying ranges of diffusion timesteps during inference. The visualization highlights that earlier timesteps predominantly contribute to semantic aspects, whereas the later timesteps are instrumental in capturing fine-grained details.

## 4.2    RANDOMIZED TRUNCATED BACKPROPAGATION

During our experimentation, we encountered a significant issue with full backpropagation through time (BPTT) - it led to mode collapse within just two training epochs. Irrespective of the input conditioning prompt, we observed that the model consistently generated the same image. To address this challenge, we explored truncated backpropagation through time (TBTT) as an alternative strategy. However, TBTT introduces a bias towards short-term dependencies, as it restricts the backpropagation to a fixed number of steps, denoted as $K$ (a hyperparameter). This bias can affect gradient estimates and hinder the model's ability to capture long-range dependencies effectively. (Tallec & Ollivier, 2017) demonstrated that the bias introduced by truncation in the backpropagation through time algorithm can be mitigated by randomizing the truncation lengths, i.e., varying the number of time-steps for which backpropagation occurs.

Our human evaluation experiments, detailed in Section 5, provided valuable insights. It was observed that setting $K \sim \text{Uniform}(0, 50)$ yielded the most promising results in terms of aligned image generation. To delve deeper into the impact of different values of $K$ on alignment and image generation, we conducted a comprehensive investigation, the results of which are presented in Figure 3. Finally, the loss which we optimize is:

$$\widehat{\nabla}_\theta \mathcal{L}_{\text{TBTT-align}} = \frac{\partial \mathcal{L}_{\text{align}}}{\partial \theta} + \sum_{k=0}^{K} \frac{\partial \mathcal{L}_{\text{align}}}{\partial x_k} \cdot \frac{\partial x_k}{\partial \theta}. \tag{4}$$

## 5 EXPERIMENTS

In this study, we subject AlignProp to rigorous evaluation by employing it to fine-tune pre-trained text-to-image diffusion models across a diverse spectrum of downstream reward functions. For all our experimental scenarios, we utilize StableDiffusion (Rombach et al., 2022) a state-of-the-art pre-trained model. Our investigation is motivated by the pursuit of answers to the following questions:

- How does AlignProp compare with existing state-of-the-art alignment methods concerning ability to optimize downstream reward, data and compute efficiency?
- To what extent does the fine-tuned model exhibit generalization capabilities when presented with novel prompts?
- How does AlignProp measure up against existing state-of-the-art alignment methods in terms of human evaluations assessing the quality of the generated images?

**Baselines.** We compare against the following methods: **(a) Stable Diffusion**, the pre-trained diffusion model without any finetuning. We use Stable Diffusion v1.5 for all our experiments, **(b) Reward Weighted Regression (RWR)** (Lee et al., 2023), which uses Reward weighted negative likelihood for finetuning. Specifically, they weigh the diffusion loss with the corresponding reward for each example. Unlike AlignProp, they do not use gradients from the reward model, **(c) Denoising Diffusion Policy Optimization (DDPO)** (Black et al., 2023) current state-of-the-art method for fine-tuning text-to-image models using reinforcement learning (PPO). We use their official pytorch codebase for reproducing results, **(d) ReFL**, which is a finetuning method proposed in ImageRewards (Xu et al., 2023). where instead of doing all the denoising steps (which is about 50 for us), they do a small number of denoising steps (30-40) using DDPM scheduler. Then they do a single step of backpropagation (K=1) to update the diffusion U-Net weights. We instead do all the sampling steps as reducing the number of denoising steps reduces the fidelity of the image. Further instead of doing a single backpropagation step (K=1), we use randomized TBTT where $K \sim \mathrm{Uniform}(0, 50)$, we find this to significantly improve the generation quality as shown in Table 3 and Figure 5.

**Reward models.** For aligning text-to-image diffusion models, we explore various reward functions to guide the training process. These reward functions encompass

**(i) Aesthetics Reward**. We employ the LAION aesthetic predictor V2 (Schuhmann, 2022), which leverages a multi-layer perceptron (MLP) architecture trained atop CLIP embeddings. This model's training data consists of 176,000 human image ratings, spanning a range from 1 to 10, with images achieving a score of 10 being considered art pieces. To ensure consistency with previous research, our training procedure closely follows the approach utilized in DDPO, which employs 50 animal names as prompts.

**(ii) Human Preference Reward**. We use Human Preference v2 (Wu et al., 2023a), where CLIP model is fine-tuned using an extensive dataset comprising 798,090 human ranking choices across 433,760 pairs of images. Notably, this dataset stands as one of the largest of its kind, facilitating robust image-text alignment. For our training regimen, we draw from the 2400 prompts introduced in the HPS v2 dataset.

**(iii) Concept Removal**. This is focused on the removal of specified object categories from generated images. This task holds the potential for curbing the generation of abusive or harmful content by text-to-image models. In this instance, we address a simplified version of the task, instructing the text-to-image model to generate images containing "<Concept Name> and Books." To facilitate this task, we enlist the aid of OwlViT (Minderer et al.), an open vocabulary object detector capable of identifying "books" within images. As our reward function, we adopt a confidence score-based approach, specifically (1.0 - $c$), where $c$ represents the confidence score assigned by OwlViT to its "books" detection.

### 5.1 SAMPLE AND DATA EFFICIENCY IN REWARD FINETUNING

The task of pre-training foundation models, such as Stable Diffusion, on large-scale training datasets is currently within the reach of only a select few entities with access to substantial computing resources. However, the subsequent fine-tuning of these foundation models towards downstream objectives should ideally be accessible to a broader range of entities, including academic institutions with more

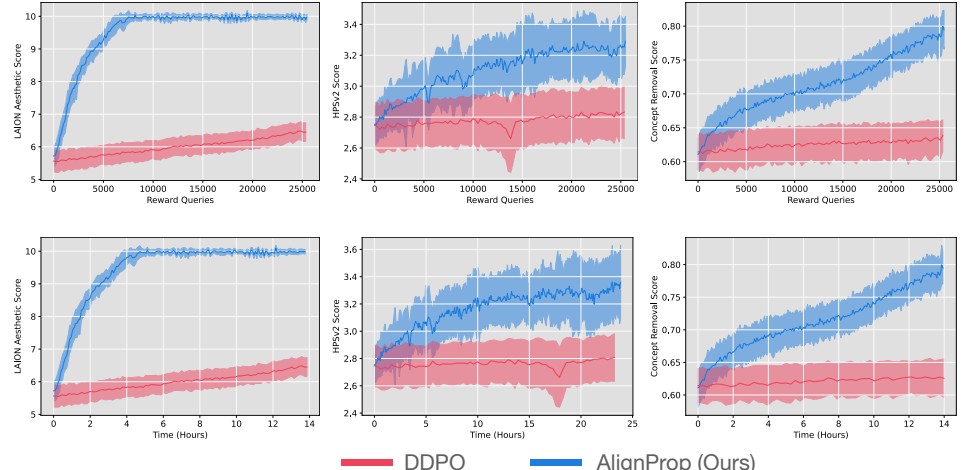

Figure 4: Reward finetuning results on multiple reward functions. In the top half of the figure, we compare the data efficiency of AlignProp and DDPO. In the bottom half of the figure we compare the convergence speed of AlignProp and DDPO. As seen AlignProp outperforms DDPO on both metrics.

limited computational capabilities. To enable such accessibility, it is imperative that an effective fine-tuning method for foundation models be both sample-efficient and incur low computational costs.

In this section, we undertake a comprehensive comparison between AlignProp and DDPO () across various reward functions, examining the achieved rewards and the computational and data efficiency. Our results are visually presented in Figure 4. Notably, our method demonstrates superior performance in terms of rewards while simultaneously exhibiting remarkable data efficiency, being 25 times more data-efficient than DDPO.

To further emphasize the efficiency of our approach, we conduct a comparative analysis of wall-clock times for DDPO and AlignProp on identical computational resources, which include 4 A100 GPUs and the same batch size. As illustrated in the HPS v2 dataset scenario, AlignProp achieves a score of 2.8 in just 48 minutes, whereas DDPO requires approximately 23 hours, highlighting a substantial 25-fold acceleration in convergence speed. Additionally, we provide qualitative results in Figure 5, offering a visual representation of the outputs generated by both AlignProp and DDPO. Notably, our method excels in generating significantly more artistic images when compared to DDPO.

| Method | Animals | | HPS v2 | |
|---|---|---|---|---|
| | Train. | Test. | Train. | Test. |
| Stable Diffusion | 5.73 | 5.64 | 2.74 | 2.86 |
| RWR | 6.21 | 6.04 | 2.81 | 2.91 |
| DDPO | 7.18 | 6.82 | 2.87 | 2.93 |
| AlignProp (Ours) | **8.94** | **8.71** | **3.30** | **3.32** |

Table 1: **Reward on novel text prompts** (higher is better). We split the class names in these datasets into train and test split such that there is no overlap in them. The proposed method AlignProp achieves higher reward compared to all other baselines.

## 5.2 GENERALIZATION TO NEW TEXT PROMPTS

One pivotal consideration favoring model finetuning, as opposed to prompt finetuning () or initial noise $x_T$ finetuning (), is the model's potential for generalization to new prompts. Here, we evaluate the finetuned models stemming from AlignProp and baseline approaches, specifically assessing their capacity for generalization.

Here, we consider two different prompt datasets, each with a different type of split: (a) **Animals:** We start with a dataset encompassing a curated list of 50 animals, mirroring the training set configuration employed in DDPO. However, in the test set, we introduce a novel set of animals as prompts that were not encountered during the training phase. The training process in this scenario is conducted using the Aesthetic reward model, (b) **HPS v2**: The Human Preference dataset () offers a diverse array of 3200 prompts, categorized into four different styles. Each style comprises roughly 800

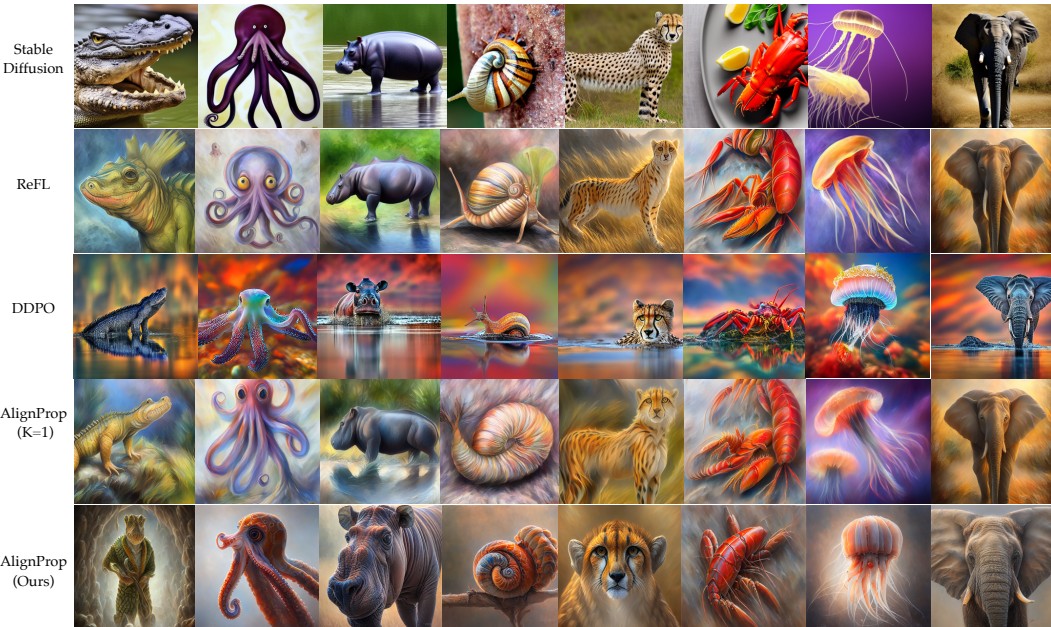

Figure 5: Qualitative comparison with baselines is conducted on novel animals that were not encountered during the training phase. In this scenario, both AlignProp and the baselines are trained using an Aesthetic reward model.

prompts. To establish a train-test split, we adopt a random selection approach, allocating 600 prompts for training and reserving 200 for testing. In this case, we utilize the HPS-v2 reward model for the training regimen.

In Table 1, we present the average rewards achieved by the models on both the train and test splits after 100 epochs of training. Notably, AlignProp consistently outperforms the baseline methods in terms of the generalization exhibited by the resulting fine-tuned models.

### 5.3 HUMAN EVALUATIONS FOR FIDELITY AND IMAGE-TEXT ALIGNMENT

We conducted a human preference study using Amazon Mechanical Turk, performing a paired comparison test between the proposed method and different baselines such as ReFL, DDPO, and Stable Diffusion. In this evaluation, we employed the HPS reward function. For the assessment of image-text alignment, we presented two image generations from each method (ours and the baseline) using the same prompt, accompanied by the question: 'Which image is more consistent with the text?' In terms of fidelity, we displayed two samples for a specific prompt from both the baselines and AlignProp, with the question: 'Which image quality appears better?' We collected 500 responses for each task. As depicted in Table 2, the proposed method is preferred over the baselines. For additional qualitative results on HPS reward function and prompts, please refer to Appendix Figure 7

### 5.4 MIXING WEIGHTS

In this context, we demonstrate the ability of AlignProp to interpolate between different reward functions during the inference phase. We draw inspiration from the concept presented in ModelSoup (Wortsman et al., 2022), which showcases how averaging the weights of multiple fine-tuned models can enhance image classification accuracy. Expanding upon this idea, we extend it to the domain of image editing, revealing that averaging the LoRA weights of diffusion models trained with distinct reward functions can yield images that satisfy multiple reward criteria.

To achieve this, we employ a straightforward approach: suppose $\theta_1$ and $\theta_2$ represent the LoRA weights trained using reward functions $R_1$ and $R_2$ respectively. In this parameter space, we perform interpolation by computing a weighted average as $\theta_{\text{mod}} = \alpha \cdot \theta_1 + (1 - \alpha) \cdot \theta_2$ where $\alpha$ is a scalar ranging from 0 to 1.

To better understand the effect of mixing we consider two complementary reward functions specifically compressibility and aesthetics. For compressiblity we finetune CLIP supervised using the JPEG compression reward from (Black et al., 2023). As illustrated in Appendix Table 4 and Figure 6, AlignProp adeptly demonstrates its capacity to interpolate between distinct reward functions, achieving the highest overall reward when the mixing coefficient is set to 0.5.

| Method | Fidelity | Image-text Alignment |
|---|---|---|
| Stable Diffusion | 20.8% | 34.4% |
| AlignProp (Ours) | **79.2**% | **65.6**% |
| ReFL | 28.8% | 35.0% |
| AlignProp (Ours) | **71.2**% | **65.0**% |
| DDPO | 17.6% | 36.4% |
| AlignProp (Ours) | **82.4**% | **63.6**% |

Table 2: We showcase Human Evaluation Scores, with the percentage indicating the frequency with which humans preferred the generated images from AlignProp over the baselines

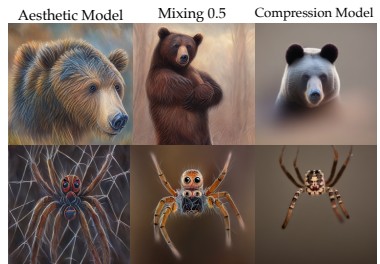

Aesthetic Model    Mixing 0.5    Compression Model

Figure 6: Here, we depict the image generations produced by the Aesthetic Model, Compression Model, and a hybrid model formed by averaging the parameters of both. The mixing coefficient for this combined model is set at 0.5

**Ablations:** In Table 3, we analyze various design choices made throughout our experiments. We do our ablations while using HPSv2 as our reward function. Firstly, we explore different choices for 'K' in truncated backpropagation. Subsequently, we investigate the impact of using EDICT affine coupling layers to reduce computational memory. EDICT, as described in (Wallace et al., 2023), revises the forward and backward diffusion processes through affine coupling layers. However, we observed that EDICT layers significantly slow down the process (by a factor of 2) compared to simple gradient checkpointing and also yield subpar performance. Additionally, we evaluate the choice between LoRA and the complete UNet fine-tuning strategy. Finally, we assess different scheduling methods, comparing DDIM with Ancestral sampling.

## 6 LIMITATIONS

AlignProp leverages gradients derived from the reward function to finetune diffusion models enhancing both sampling efficiency and computational efficacy. However, this reliance on a differentiable reward function introduces a caveat. In instances where the reward function is imperfect, the fine-tuning process may lead to over-optimization, deviating from the intended outcome. Addressing and mitigating this risk of over-optimization stands as a direct focus for future research.

| | Avg. Reward |
|---|---|
| AlignProp (Ours) | **3.30** |
| with K=1 | 2.91 |
| with K=10 | 3.14 |
| with EDICT | 2.95 |
| w/o LoRA | 3.09 |
| w/o DDIM | 2.97 |

Table 3: We compute the average reward for both the proposed method, AlignProp, and the baseline models after completing 100 training epochs. The table demonstrates that the proposed method consistently outperforms all other baselines in terms of achieved rewards.

## 7 CONCLUSION

We have introduced AlignProp, a differentiable framework for fine-tuning pre-trained diffusion models, optimizing directly across a spectrum of diverse reward functions. By conceptualizing the denoising process as a recurrent, differentiable policy responsible for mapping noise and prompt-conditioning information to output images, we have introduced a class of pathwise gradient algorithms that surpass existing reinforcement methods. To address concerns related to over-optimization, AlignProp incorporates truncated backpropagation and manages memory requirements through low-rank adapter modules and gradient checkpointing. Our experiments consistently highlight AlignProp's efficacy in optimizing diverse reward functions, even for tasks that prove challenging to specify through prompts alone. Looking ahead, future research avenues may explore the extension of these concepts to diffusion-based language models, with the aim of enhancing their alignment with human feedback.

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

# A APPENDIX

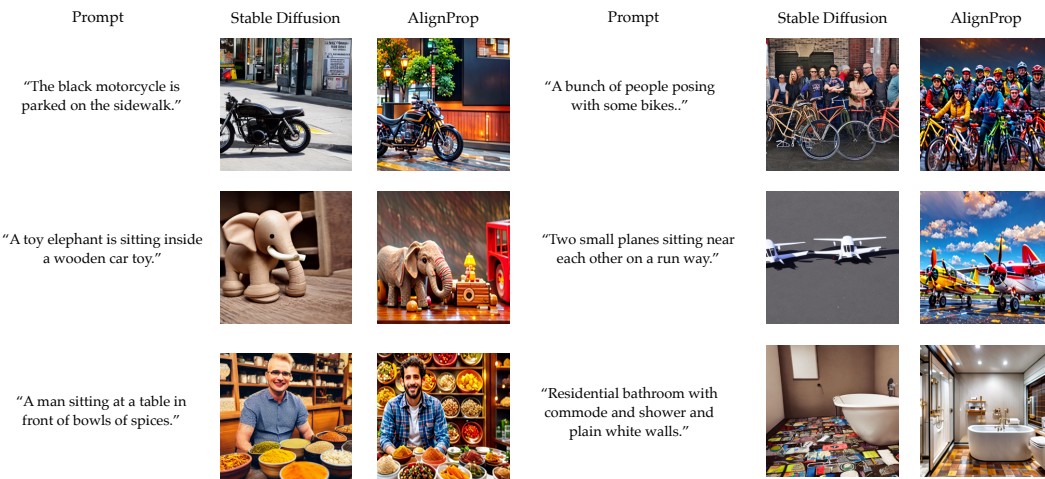

Figure 7: In this Figure we qualitatively compare AlignProp with Stable diffusion while using multiple prompts of HPS v2 evaluation prompt dataset. As can be seen AlignProp achieves higher fidelity results with better image-text alignment.

|  | Aesthetic | Compressibility | Aesthetic + Compressibility |
|---|---|---|---|
| Compressibility Model | 5.32 | **8.32** | 6.82 |
| Mixing 0.5 | 6.61 | 7.92 | **7.27** |
| Aesthetic Model | **7.13** | 5.57 | 6.35 |

Table 4: In the above Table, we compare the mixing results between Compressibility model and Aesthetic Model. As can be seen from the results, mixing the weights achieves the best results when the metric accounts for an average of both the scores. Additionally it achieves second best results on the individual metrics, thus indicating that mixing does result in an interpolation in the reward space.

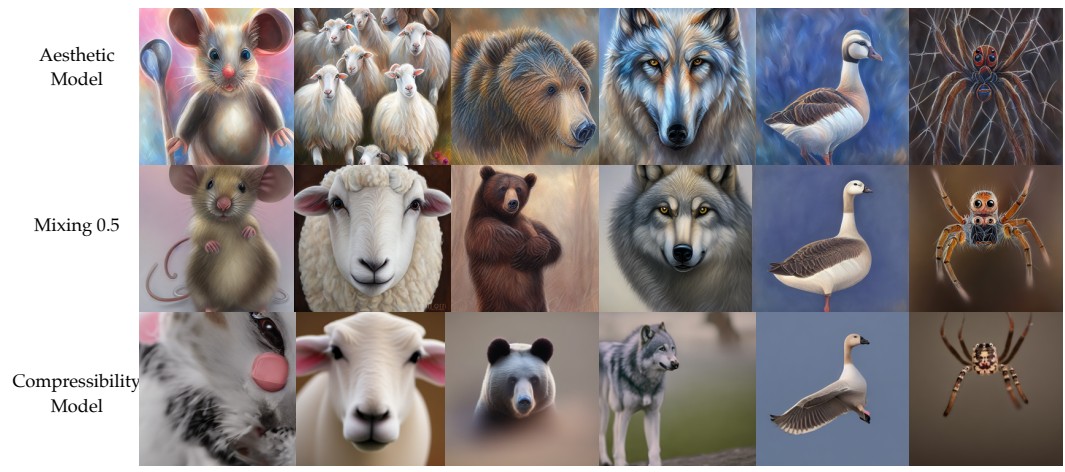

Figure 8: In this Figure we show additional mixing results between compressibilty and Aesthetic model.

**Implementation details** We use 4 A100 GPUs for all our experiments, with Adam Optimizer and a learning rate of 1e-3. We build our codebase on top of Pytorch 2.0. Few reward functions such as

aesthetics that are not text-conditioned and are trained on relatively small datasets can result in a loss of image-text alignment if optimized above a specific score value. To prevent this we do early stopping (10th epoch in the training process).

