# OpenReview forum: "Aligning Text-to-Image Diffusion Models with Reward Backpropagation"
_ICLR.cc/2024/Conference — Submitted to ICLR 2024_

### Official Review · Reviewer_VMQ3 · 2023-10-24

**Soundness:** 2 fair
**Presentation:** 2 fair
**Contribution:** 2 fair
**Rating:** 3
**Confidence:** 4

**Summary:**

The paper introduces AlignProp, a novel method for aligning text-to-image diffusion models with specific reward functions. The authors argue that existing reinforcement learning methods are inefficient due to high variance gradients. AlignProp aims to overcome these limitations by using end-to-end backpropagation to fine-tune the model. The paper also discusses techniques to manage memory overhead, such as fine-tuning low-rank adapter (LoRA) modules and using gradient checkpointing.

**Strengths:**

Comparative Analysis: The paper does a good job of positioning AlignProp against existing methods, particularly reinforcement learning techniques. This helps in understanding the unique advantages of AlignProp.

Results: The paper claims that AlignProp achieves higher rewards in fewer training steps and is preferred by human users, although the empirical evidence supporting these claims could be strengthened.

**Weaknesses:**

* The writing is not good. \cite and \citep are different. And there are many other typos and missing/wrong references, which cause some difficulties in understanding the work.
* The idea is simple and straightforward. If authors want to convince me to increase the score and demonstrate the effectiveness of the approach,  can authors provide an anonymous  website to list un-cherry-picked images across different iterations and different methods? It is good to use open-eval prompts, such as the ones provided in part image generation eval benchmark.

**Questions:**

See above

---

> ### Author Response · Authors · 2023-11-29
> **Response to Reviewer VMQ3**
>
> **Q4.1) Typos**
> Thanks for pointing this, please check our updated pdf, we believe we have
> fixed all of them.
>
> **Q4.2) If authors want to convince me to increase the score and
> demonstrate the effectiveness of the approach, can authors provide an
> anonymous website to list un-cherry-picked images across different
> iterations and different methods?**
> **Aesthetics reward model:**
>
> In the following webpage
> [<link1>](https://wandb.ai/alignprop-iclr/alignprop/reports/Aesthetics-Results-different-Epochs---Vmlldzo2MDU5MzI4?accessToken=vph90cg4qa44h0dl15nbxfm4tu9b3efv6c55r8ssf39yv3e9t3mzon82q9eq221j),
> we show results for different epochs of training for our model and the
> baselines. As can be seen in the visuals, our model achieves better
> qualitative results even in the intermediate epoch of training.
>
> **HPS reward model:**
>
> In the following webpage
> [<link2>](https://alignprop-iclr.github.io/hps.html), we compare AlignProp
> with baselines over  unseen prompts from the evaluation prompt set of HPS
> reward model.
>
>
> **Disabling LoRA weights:**
>
> In the following webpage
> [<link3>](https://alignprop-iclr.github.io/index.html), we expand Figure 3
> of our paper and show more example cases for dropping lora weights over
> different timesteps of denoising.

---

### Official Review · Reviewer_e5M5 · 2023-10-30

**Soundness:** 2 fair
**Presentation:** 3 good
**Contribution:** 1 poor
**Rating:** 5
**Confidence:** 4

**Summary:**

This paper aims at aligning a pre-trained text-to-image diffusion model with downstream objectives using the most straightforward way. The proposed AlignProp fully reconstructs the input images during training, which are then taken as input to an off-the-shelf reward model for end-to-end reward optimization. The authors utilize several well-acknowledged optimization tricks for better GPU memory management and aligning efficiency. The extensive experiments demonstrate the superiority of AlignProp.

**Strengths:**

- The authors construct this paper with clear architecture and detailed discussion.
- The proposed method is fairly simple with extensive experiment results.
- The motivation is relatively clear and consistent with the human intuition.

**Weaknesses:**

- Novelty would be a controversial problem of this paper:
  - Methodology: The main technical components of this paper consist of two parts: 1) directly propagating the reward back to the diffusion models without RL, following DDPO, and 2) spanning the whole denoising process and perform back propagation through time, which has been utilized for diffusion model guidance / alignment early in DiffusionCLIP published in CVPR 2022.
  - Implementation: According to the authors, the main difficulty of applying the methodology above in the considered datasets is the high GPU memory usage, which, however, can be simply solved by several commonly used optimization tricks of diffusion models including LoRA, gradient checkpointing and gradient truncation.
  - Therefore, although nobody has conducted experiments by combining all these things together before, it is still hard to convince me that there exists a novel research problem and this work should be considered as a research paper for top-tier conference like ICLR instead of a solid technical report.
- About compute efficiency
  - In order to propagate gradient back to the whole sample chain, AlignProp needs to perform the whole T (=50 according to the authors) for each data sample during training even with DDIM, which instead requires only 2 steps for DDPO, and this is a huge efficiency gap.
  - Therefore, I cannot understand why in the 2nd row of Fig. 4, AlignProp can demonstrate efficiency even with respect to Time. Can you give more implementation details about how these experiments are conducted, and why the re-implemented DDPO converges so quickly?
  - Another perceptive to view DDPO and AlignProp together is that both of them are working on a practical estimation towards fully fine-tuning UNet with respect to reward models using BPTT. DDPO chooses to back propagate only one step, while AlignProp chooses to use LoRA and gradient checkpointing.
  - Therefore, a more fair comparison should be between AlignProp and DDPO with fully fine-tuning and no gradient checkpoint.

- About generalization of the proposed AlignProp:
  - I wonder if AlignProp can generalize to any circumstances as long as there exists a reward model.
  - In other words, if the alignment ability requires the pre-trained diffusion models have the downstream-desired generation ability at first (e.g., Aesthetics). For example, can AlignProp be applied to allow Stable Diffusion to generate medical images with a reward model trained on medical images?
  - In the 2nd paragraph, the authors claim that the motivation to utilize reward models instead of supervised fine-tuning is the requirement of high-quality data samples. However, in my understanding even aligning with reward models still require these high-quality samples with high rewards for alignment optimization.

**Questions:**

- Implementation details:
  - During fine-tuning, are the reward model parameters $\phi$ fixed or also tunable?
  - What is the specific T value (i.e., total denoising steps) utilized in your experiments? According to the "Baselines" paragraph in Sec. 5, the T value is set to be 50, which, however, is 1000 for Stable Diffusion trained with DDPM during pre-training.
- Writing:
  - Fix the citation style by using the correct latex command.
  - Typo in the 2nd line of Page 5 ($\pi_{\theta}$ instead of $\pi_theta$)

---

> ### Author Response · Authors · 2023-11-29
> **Response to Reviewer e5M5**
>
> **Q3.1) Paper lacks novelty.**
>      **Please, see the general response.**
>
>
> **Q3.2) Comments on Diffusion CLIP.**
> Thanks for the reference! We will cite Diffusion CLIP, we were not aware
> of the work. As per the differences, DiffusionCLIP focuses on image
> translation, that is mapping images a dataset of source images to target
> domain, while using domain-specific small-scale diffusion models. We on
> the other hand do not require any source images and focus on aligning
> large-scale foundational text-to-image models.  **Please, also see the
> general response.**
>
>
> **Q3.3) Compute Efficiency. Fair comparison should be between AlignProp
> and DDPO with fully fine-tuning and no gradient checkpoint.**
>
> We use the official pytorch repository of DDPO for all our experiments.
> DDPO (in their updated version) uses LoRA adapter weights. AlignProp
> however also uses gradient checkpointing while DDPO doesn’t, note that we
> only use gradient checkpointing to fit our model into GPU memory. Gradient
> checkpointing trades off compute with memory thus making AlignProp slower.
> Including gradient checkpointing in DDPO will only make its speed worse
> without any benefit, therefore we don't run this ablation. As per the
> perspective you mentioned, DDPO should indeed be 50x faster than AlignProp
> as it does a single step of backprop while we do 50 steps of
> backpropogation. However in DDPO's implementation they accumulate
> gradients over all denoising timesteps, to make the noisy gradients more
> stable. As can be seen in their code
> [here](https://github.com/kvablack/ddpo-pytorch/blob/1958463f020112c9a7bc85768d296daacc2e1b4b/scripts/train.py#L501),
> this makes learning from N samples 50x slower. Further they have an
> additional sampling loop which generates RGB images to obtain it's reward.
> We on the other hand don't have a seperate sampling and training loop.
> Overall to learn from 128 samples, DDPO takes 1 minute 20 seconds to
> sample them, then it takes 8 minute 10 seconds to train on them. We on the
> other hand take 6 minute 16 seconds to sample and learn from 128 samples.
>
> **Q3.4)  Can AlignProp generalize to any circumstances as long as there
> exists a reward model:**
> We think farther the reward model from the training distribution of Stable
> Diffusion the more sample inefficient it becomes to tune it. To study this
> hypothesis, we ran a toy experiment where we took 10 random images from
> [CT Medical
> Images](https://www.kaggle.com/datasets/kmader/siim-medical-images)
> dataset, and 10 random images from ImageNet. Our reward model is simply
> the similarity with the best matching image from the 10 examples.
> Emperically, we find that AlignProp achieves similar reward on both
> datasets, however it ends up being 60x more sample efficient on ImageNet
> than the medical dataset.
>
>
> **Q3.5) Why not supervised finetuning? Aligning with reward models still
> require these high-quality samples.**
>
>
> Reward models are trained by humans ranking **images generated by the
> generative model, therefore the process does not require access to any
> real-world high-quality  samples**. Supervised fine-tuning on the other
> hand requires access to real-world high-quality samples. This is a
> central benefit of  reward-guided alignment.
>
> **Q3.6) During fine-tuning, are the reward model parameters  fixed or also
> tunable?**
> The reward models are  fixed.
>
> **Q3.7)  T value is set to be 50, which, however, is 1000 for Stable
> Diffusion**
> We use DDIM sampling which reduces the sampling steps from 1000 to 50 by
> making the generation process deterministic and non-markovian.
>
> **Q3.7) Mentioned Typos**
> Thanks you, we will fix these typos.

---

### Official Review · Reviewer_JeN1 · 2023-10-31

**Soundness:** 3 good
**Presentation:** 3 good
**Contribution:** 2 fair
**Rating:** 5
**Confidence:** 4

**Summary:**

This paper introduces "AlignProp," a technique that refines text-to-image diffusion models using end-to-end backpropagation of the reward gradient during the denoising phase. Rather than consuming excessive memory, AlignProp fine-tunes low-rank weight modules and uses gradient checkpointing. When tested, AlignProp efficiently optimized diffusion models for various objectives like semantic alignment and aesthetic enhancement. It outperformed existing methods, achieving better results in fewer steps, making it a preferred choice for optimizing diffusion models.

**Strengths:**

The experiments conducted are comprehensive and of high quality.

**Weaknesses:**

1. **Originality & Novelty:** The paper seems to lack significant originality and novelty. Implementing the two memory-saving techniques - finetuning with LoRA and gradient checkpointing - does not appear challenging, especially since they are already available in the “diffusers” package. Further, randomizing the number of denoising steps appears to be a straightforward approach, and it's not guaranteed to address the collapsing issue.

2. **Previous Work Reference:** The concept of using a differentiable reward function and backpropagating the gradient directly to each timestep was earlier introduced in the Diffusion-QL[1] paper. It would be beneficial to cite the D-QL paper. There's also another very recent concurrent work on the subject [2].

3. **Issue of Collapsing:** Both the present paper and [2] have highlighted the issue of collapsing when using a differentiable reward function for backpropagation. As demonstrated in the DDPO paper, using policy gradient through a non-differentiable reward signal doesn't present this issue. A more detailed exploration of the collapsing issue, along with effective mitigation strategies, would be a valuable addition. The currently proposed method of randomized denoising length seems somewhat simplistic. A more robust solution, accompanied by a thorough analysis, is anticipated.

[1] Wang, Zhendong, Jonathan J. Hunt, and Mingyuan Zhou. "Diffusion policies as an expressive policy class for offline reinforcement learning." arXiv preprint arXiv:2208.06193 (2022).

[2] Clark, Kevin, et al. "Directly Fine-Tuning Diffusion Models on Differentiable Rewards." arXiv preprint arXiv:2309.17400 (2023).

**Questions:**

1. In Figure 4, the HPSv2_Score is noted as ranging from 2.4 to 3.6. However, to the left of Table 1, the paper mentions a score of 0.28. Is this a typographical error?

2. In Figure 5, the color map and object shapes from AlignProp appear quite similar. Yet, the results from Stable Diffusion and DDPO exhibit greater diversity. Could you clarify this?

---

> ### Author Response · Authors · 2023-11-29
> **Response to Reviewer JeN1**
>
> **Q2.1) Paper lacks significant novelty.**
>     **Please, see the general response.**
>
>
> **Q2.2) Randomizing the number of denoising steps  not guaranteed to
> address the collapsing issue.**
>
> In all our experiments across all reward models we tried, we consistently
> find randomizing the denoising steps to help prevent the collapsing issue.
>
> **Q2.3) Comments on Diffusion-QL[1] and concurrent work of DRAFT[2]**
>
> Thanks for the reference! We will cite DQL, we were not aware of the work.
> As per the differences, DQL focuses on offline reinforcement learning for
> robotics tasks, we on the other hand focus on aligning large-scale
> foundational text-to-image models.
>
> DRAFT[2] indeed is a concurrent work , it in fact came out a week after
> the ICLR paper submission deadline. Having said that, they still don’t do
> full backpropagation through time, which might prevent them from easily
> tuning at a semantic level, as they don’t update the earlier steps of
> denoising.
>
> **Q2.4) Issue of Collapsing, DDPO doesn't collapse.**
>
> Yes indeed DDPO doesn’t collapse with full back propagation. We think
> policy gradient methods are inherently regularized due to noise from the
> credit assignment issue thus don’t end up collapsing. However the noise
> also results in sample inefficiency as we show in our results. We instead
> do direct reward backpropagation while keeping the full backprop of DDPO
> by randomizing the truncation steps.
>
> **Q2.5) Typographical error with HPS score in text**
> Yes this is an error, thanks for pointing. We meant 2.8.
>
> **Q2.6) In Figure 5, lack of color map and object shapes diversity in
> AlignProp, whereas Stable Diffusion and DDPO show greater diversity.**
>
>
> We think this has to do with the training set of the reward model.  For
> aesthetics reward models, highest-rated images mostly contain portraits or
> artworks. Optimizing the reward model thus results in a mode collapse
> where the model generates images with similar color map and object poses
> (facing the camera). Stable Diffusion doesn’t see such a collapse as it’s
> not trained on this reward function but rather to model a much diverse
> data distribution. **DDPO does see a type of collapse, as can be seen in
> Figure 4 where the images generated by DDPO are mainly in the water
> background.  We think it doesn’t get the exact same form of collapse as
> our model as it’s underfitting to the reward function,** due to noisy
> policy gradient training. Further, we don’t see such a collapse while
> training on HPS reward function as can be found here
> **[link](https://alignprop-iclr.github.io/hps.html) and Figure 7** in the
> supplementary, we think this is because HPS reward function is trained
> using a much diverse dataset.

---

### Official Review · Reviewer_MYQY · 2023-11-01

**Soundness:** 3 good
**Presentation:** 3 good
**Contribution:** 2 fair
**Rating:** 6
**Confidence:** 3

**Summary:**

In this paper proposes AlignProp, a method that aligns diffusion models to downstream reward functions using end-to-end backpropagation of the reward gradient through the inference chain. The main challenge the paper is trying to solve is the prohibitive memory cost required by naive implementation of such backpropagation.

**Strengths:**

- The paper studies an important problem of end-to-end backpropagating a reward function through the denoising process.
- The presented results look promising and the experiments are extensive and convincing.

**Weaknesses:**

- Clarification: in eq 3, does the first term come from weight decay?
- Typos: 1) eq 3 and 4, cdot notations are not consistent; 2) page 5 "policy \pi_{theta}"; 3) page 5 "k"m".
- Figure 3 presents visual results on a single image, which seems not "comprehensive" enough to study the impact of value of K (as stated in the last paragraph in page 5).

**Questions:**

- It might be interesting to study other PEFT methods besides LoRA.

---

> ### Author Response · Authors · 2023-11-29
> **Response to Reviewer MYQY**
>
> **Q1.1) Does the first term in Eqn 3 come from weight decay?**
> Yes, the first term is a result of the weight decay loss.
>
> **Q1.2) Might be interesting to study other PEFT methods besides LoRA.**
> Thanks for the suggestion, we tried using
> [IA3](https://arxiv.org/abs/2205.05638), we indeed get some improvement in
> performance. For instance with Aesthetic reward function, AlingProp is
> about 2% more sample efficient, we will include this result in our final
> submission.
>
> **Q1.3) Typos**
>  Thanks for pointing out we have fixed these typos.
>
> **Q1.4) More examples for Figure 3 to study the impact of value of K**
>
> Thanks for the suggestion, you can find more examples in the following
> [link](https://alignprop-iclr.github.io/index.html)

---

### Author Response · Authors · 2023-11-29
**General Response**

Dear Reviewers,

Thank you for your  comments.

Many Reviewers  brought up lack of novelty as the main weakness of the
paper.

**Contribution:** AlignProp addresses the task of aligning large-scale
foundational text-to-image models using differentiable reward functions,
by end-to-end backpropagation of reward gradients to diffusion model
parameters. The state-of-the-art method for this task till our work was
DDPO that uses policy gradients, **and thus does not use the gradients of
the reward function**. Our method directly backpropagates gradients from
the reward model to update the diffusion model.

**Reviewer e5M5 says: "directly propagating the reward back to the
diffusion models without RL, following DDPO" We believe this is
incorrect:** there is a significant difference between the two methods,
DDPO does not backpropogate gradients from the reward model (it does not
use the gradient of the reward function, just its values in specific image
samples), whereas we do.


**Closest related works:** The closest related work to AlignProp is
DiffusionQL and DiffusionCLIP, as also pointed out by reviewers JeN1 and
e5M5?. While all the above mentioned methods use reward backpropogation to
update the diffusion model, the use-cases are very different which results
in very different set of challenges for the three works.

DiffusionQL[1] addresses the problem of offline reinforcement learning
where a small-scale policy neural network is trained via Q-learning to
generate high rewarding robot actions. AlignProp instead focuses on
aligning billion parameter  large-scale  pretrained text-to-image models,
a very different setup and computational challenges.


DiffusionCLIP[2] is a category-specific image translation model, which
maps a set of source images (male face images) to a target domain
(female). It does so by doing DDIM inversion on the source images to
obtain latents z, it then updates the denoising process such that the
generated image matches with the text by backpropagating gradients from a
text-image matching CLIP objective. It uses unconditional diffusion models
trained on category-specific datasets such as Dogs or faces. We on the
other hand update large-scale text-conditioned diffusion models such as
Stable Diffusion.  Instead of doing DDIM inversion using a dataset of
source images, we directly generate the images from the foundation
diffusion model given a set of prompts.  As one can see, these setups are
very different, they address very different problems with different
methodologies, and pose unique challenges for each, as well as very
complementary benefits to the community.

Reviewer JeN1 brought up DRAFT[3], which is a concurrent work, and in fact
it came out a week after the ICLR paper submission deadline. **If
AlignProp was indeed obvious, we do not see why a work such as DRAFT would
exist.** Further, unlike AlignProp, DRAFT does not adapt earlier denoising
steps  which might prevent them from easily adapting at a semantic level,
as we show in Figure 3 of our paper.


Reviewer JeN1 says: "finetuning with LoRA and gradient checkpointing -
does not appear challenging, especially since they are already available
in the “diffusers” package." You are correct.  However, **we do not think
a method should be evaluated based on the number of line changes in the
code, but based on how valuable of a contribution it makes in an important
problem.**  There are many useful works in the past such as beta-VAE, that
have mainly changed  a single hyperparameter value and still been very
useful to the community.

**Additional Dense Qualitative results:**

**Aesthetics reward model:**

In the following webpage
[<link1>](https://wandb.ai/alignprop-iclr/alignprop/reports/Aesthetics-Results-different-Epochs---Vmlldzo2MDU5MzI4?accessToken=vph90cg4qa44h0dl15nbxfm4tu9b3efv6c55r8ssf39yv3e9t3mzon82q9eq221j),
we show results for different epochs of training for our model and the
baselines. As can be seen in the visuals, our model achieves better
qualitative results even in the intermediate epoch of training.

**HPS reward model:**

In the following webpage
[<link2>](https://alignprop-iclr.github.io/hps.html), we compare AlignProp
with baselines over  unseen prompts from the evaluation prompt set of HPS
reward model.


**Disabling LoRA weights:**

In the following webpage
[<link3>](https://alignprop-iclr.github.io/index.html), we expand Figure 3
of our paper and show more example cases for dropping lora weights over
different timesteps of denoising.

---

> ### Author Response · Authors · 2023-11-29
> **General Response**
>
> [1] Wang, Zhendong, Jonathan J. Hunt, and Mingyuan Zhou. "Diffusion
> policies as an expressive policy class for offline reinforcement
> learning." arXiv preprint arXiv:2208.06193 (2022).
>
> [2] Kim, Gwanghyun, Taesung Kwon, and Jong Chul Ye. "Diffusionclip:
> Text-guided diffusion models for robust image manipulation." Proceedings
> of the IEEE/CVF Conference on Computer Vision and Pattern Recognition.
> 2022.
>
> [3] Clark, Kevin, et al. "Directly fine-tuning diffusion models on
> differentiable rewards." arXiv preprint arXiv:2309.17400 (2023).

---

### Meta-Review · Area_Chair_SDYZ · 2023-12-12

**Metareview:**

This paper presents a method to align text-to-image diffusion models to downstream reward functions. Although  the motivation is clear and the experiments are comprehensive, most reviewers think the method is straightforward and simple, with little originality and novelty. AC agrees this work needs further extension to improve its novelty beyond a technical report.

**Justification For Why Not Higher Score:**

The method is straightforward and simple, with little originality and novelty.

**Justification For Why Not Lower Score:**

N/A

---

### Decision · Program_Chairs · 2024-01-16

Reject